# Organic Compounds as a Natural Alternative for Pest Control: How Will Climate Change Affect Their Effectiveness?

**DOI:** 10.3390/plants15010048

**Published:** 2025-12-23

**Authors:** Virginia L. Usseglio, María P. Zunino, Vanessa D. Brito, Magalí Beato, Martin G. Theumer, José S. Dambolena

**Affiliations:** 1Instituto Multidisciplinario de Biología Vegetal (IMBiV-CONICET-UNC), Av. Velez Sarsfield 1611 (X5000OPB), Ciudad Universitaria, Córdoba X5016GCN, Argentina; vusseglio@imbiv.unc.edu.ar (V.L.U.); vbrito@imbiv.unc.edu.ar (V.D.B.); mbeato@imbiv.unc.edu.ar (M.B.); 2Cátedra de Química Orgánica y Productos Naturales (FCEFyN-UNC), Av. Velez Sarsfield 1611 (X5000OPB), Ciudad Universitaria, Córdoba X5016GCN, Argentina; 3Instituto de Ciencia y Tecnología de los Alimentos (ICTA-FCEFyN-UNC), Av. Velez Sarsfield 1611 (X5000OPB), Ciudad Universitaria, Córdoba X5016GCN, Argentina; 4Centro de Investigación en Bioquímica Clínica e Inmunología (CIBIC-FCQ-UNC), Av. Medina Allende (X5000HUA), Ciudad Universitaria, Córdoba X5016GCN, Argentina; mgtheumer@unc.edu.ar

**Keywords:** biopesticides, *Fusarium verticillioides*, *Sitophilus zeamais*, global warming

## Abstract

Climate change scenarios predict increased temperatures, potentially impacting the development of phytopathogenic fungi and the efficacy of their control. This study evaluated the effects of four natural organic compounds—carvacrol, eugenol, *trans*-cinnamaldehyde, and 1-heptyn-3-ol—on the growth of *Fusarium verticillioides* and the survival of *Sitophilus zeamais* under two temperature regimes (28 °C and 32 °C). Fungal growth was assessed through the lag phase duration and mycelial expansion, while insecticidal activity was determined by mortality of *S. zeamais*. Carvacrol (1 ppm) produced the most pronounced inhibitory effect on fungal growth, significantly extending the lag phase and reducing mycelial area, with eugenol showing similar effects at selected concentrations. Both compounds maintained or enhanced their antifungal activity at elevated temperatures. Trans-cinnamaldehyde and 1-heptyn-3-ol exhibited moderate or low effects, depending on concentration and temperature. Regarding *S. zeamais*, 1-heptyn-3-ol achieved complete mortality at all concentrations under both temperature scenarios, whereas carvacrol, eugenol, and *trans*-cinnamaldehyde showed dose-dependent effects at 28 °C and enhanced efficacy at 32 °C. Overall, these findings highlight the potential of these compounds as sustainable, climate-resilient alternatives for managing fungal pathogens and stored-product pests.

## 1. Introduction

Since the Industrial Revolution, increased consumption of fossil fuels has led to significant imbalances in atmospheric composition. This disruption has contributed to the rise in global temperatures, a phenomenon recognized as a critical emergency by the United Nations Climate Change Conference (COP25) [1]. Climate change is associated with a range of environmental and biodiversity challenges, which, in turn, have profound impacts on human societies [2,3,4]. Agroecosystems are particularly vulnerable, facing reduced food security, declining crop yields, and shifts in productivity patterns [5,6]. Moreover, changing climatic conditions influence the development and adaptation of pests [7,8,9,10]. Under the Representative Concentration Pathways (RCP) 8.5 scenario, which predicts a global temperature increase of 2.5–3.5 °C above pre-industrial levels [11], crop yield losses due to insect pests are expected to double, as will losses associated with increased fungal infections and mycotoxin contamination [12,13].

The phytopathogenic fungus *Fusarium verticillioides* Sacc. (Nirenberg) is one of the major pests affecting maize production in temperate regions of the world. The presence of this fungus is associated with significant crop yield losses, ranging from 17% to 40% annually, either due to ear rot or the production of elevated levels of mycotoxins [14,15]. The presence of toxins produced by *F. verticillioides*, known as fumonisins, is correlated with the incidence of diseases in both farm animals and humans [15,16].

*Sitophilus zeamais* Motschulsky (Coleoptera: Curculionidae), the maize weevil, is one of the most important insect pests of stored grains worldwide, with a particularly severe impact in tropical and subtropical regions [17]. This insect infests maize both in the field and during storage, causing significant quantitative and qualitative losses. Its life cycle, characterized by larval development inside the kernel, hinders early detection and facilitates its spread [18,19]. In addition to direct endosperm consumption, its activity increases substrate moisture and temperature, favoring the growth of toxigenic fungi [20,21].

Traditionally, synthetic insecticides, systemic fungicides, or resistant cultivars have been used for pest control; however, these approaches are often neither fully effective nor eco-friendly [22,23,24,25]. For this reason, there is a growing search for natural alternatives for pest control. Natural organic compounds, identified from sources such as essential oils or intraspecific interactions, are being widely studied as semiochemicals of great agricultural importance [26,27,28]. However, there are few studies addressing the effectiveness of these compounds under predictive climate change scenarios.

With the increase in global temperature, mycotoxigenic fungi such as *F. verticillioides* and insect pests such *S. zeamais* have demonstrated high plasticity in adapting to climatic variations, altering their growth parameters and toxin production [29,30,31,32,33]. Nevertheless, the influence of global warming on the effectiveness of natural compounds for the control of these pests has yet to be explored. Therefore, the aim of this study was to evaluate the effectiveness of four selected organic compounds in controlling *S. zeamais* and *F. verticillioides* under global change scenarios, aiming to contribute to the design of ecologically sustainable strategies for the control of these phytopathogenic fungi.

## 2. Results

### 2.1. Carvacrol Retains Antifungal Efficacy Under Global Warming Conditions

The antifungal activity of carvacrol, eugenol, *trans*-cinnamaldehyde, and 1-heptyn-3-ol was assessed under two global warming scenarios: 28 °C (control condition) and 32 °C (RCP 8.5 scenario).

Variations in the mycelial growth rate of *F. verticillioides* under these conditions are shown in Figure 1. Control treatments (free compound) revealed that growth rate decreased on average by 1.5-fold with increasing temperature, suggesting that mycelial development may be delayed in a global warming context. This trend was consistent across most tested concentrations of the organic compounds, where growth rates at 28 °C (Figure 1A) were higher than those at 32 °C (Figure 1B).

Carvacrol exhibited the strongest antifungal activity. At higher concentrations, it almost completely inhibited the growth of *F. verticillioides* at 28 °C. Although strong antifungal activity was also observed at 32 °C, *F. verticillioides* growth rates were 5- to 14-fold lower than at 28 °C. These results indicate that the pronounced antifungal effect of carvacrol is not significantly affected by the temperature increase evaluated. Eugenol also showed significant antifungal activity at 1 ppm. The antifungal effect of eugenol was more pronounced at 28 °C, indicating that its efficacy may be compromised with increasing global temperatures.

In contrast, *trans*-cinnamaldehyde and 1-heptyn-3-ol did not show significant antifungal activity under the conditions evaluated.

Overall, these results demonstrate that the antifungal efficacy of the most bioactive organic compound would not be compromised by the temperature increases associated with climate change. In this context, compounds such as carvacrol emerge as promising natural alternatives for the control of *F. verticillioides*, even under future scenarios involving thermal stress.

### 2.2. Increased Temperature Does Not Compromise, and Enhances, the Toxicity of Volatile Compounds Toward S. zeamais

The acute effects of carvacrol, eugenol, *trans*-cinnamaldehyde, and 1-heptyn-3-ol on the survival of *S. zeamais* were evaluated under two global warming scenarios: 28 °C (control condition) and 32 °C (RCP 8.5 scenario). Mortality rates, calculated after 24 h of exposure and expressed as percentages, are presented in Figure 2. Control treatments under both scenarios showed mortality rates below 5%, indicating that the increase in temperature alone does not affect the survival of *S. zeamais*.

At 28 °C (Figure 2A), 1-heptyn-3-ol was the most effective compound for controlling *S. zeamais*, whereas carvacrol, *trans*-cinnamaldehyde, and eugenol produced lower effects. At 0.125 ppm, the highest concentration tested, all compounds achieved 100% mortality; however, this effect persisted only for 1-heptyn-3-ol when the concentration was reduced five-fold. At this reduced concentration (0.025 ppm), carvacrol and *trans*-cinnamaldehyde exhibited similar effects, reaching mortality rates of 60.00 ± 4.08% and 62.50 ± 6.29%, respectively. At the same concentration, eugenol showed a mortality rate approximately 20% lower, a difference that was statistically significant compared with the other compounds. At 0.0125 ppm, the high mortality rate induced by 1-heptyn-3-ol was maintained, whereas mortality rates for the other three compounds decreased significantly, ranging between 15% and 22%, with carvacrol showing a slightly higher value.

Under the RCP 8.5 scenario at 32 °C (Figure 2B), 1-heptyn-3-ol maintained its effectiveness, reaching 100% mortality at all three concentrations tested (0.125, 0.025, and 0.0125 ppm), whereas the other compounds exhibited increased efficacy compared with the results observed at 28 °C. Carvacrol achieved 100% mortality at both 0.125 and 0.025 ppm, and at 0.0125 ppm the mortality rate reached 92.5%. In the case of *trans*-cinnamaldehyde and eugenol, no clear dose-dependent response was observed. *Trans*-cinnamaldehyde showed the lowest mortality percentage at the highest concentration tested (0.125 ppm), although this value was not statistically different from those obtained at 0.025 and 0.0125 ppm. A similar pattern was observed for eugenol; however, in this case, the mortality rate at the highest concentration, 0.125 ppm, was 15–20% lower than those recorded for the other concentrations, and this difference was statistically significant. Despite these specific response patterns, mortality rates obtained under this global warming scenario were markedly higher than those recorded at 28 °C, indicating that, under projected increases in global temperature, the effectiveness of these natural compounds against *S. zeamais* would not be compromised and may even be enhanced.

## 3. Discussion

### 3.1. Sustained Efficacy of Carvacrol and Eugenol Against F. verticillioides Growth at Elevated Temperatures: Implications for Biocontrol on Climate Change Scenarios

Our findings reveal that the increase in global temperature resulted in a slight, non-significant reduction in fungal growth rate, suggesting that *F. verticillioides* may maintain its pathogenic potential under future climatic conditions. All tested organic compounds under simulated climate change conditions (32 °C) extended the lag phase, indicating an inhibitory effect on early adaptation and metabolic activation stages. These findings suggest that future atmospheric scenarios may influence the ecological fitness and growth rates of this phytopathogen.

Among the organic compounds tested, carvacrol exerted the most pronounced antifungal activity under simulated climate change conditions. The antifungal effect of carvacrol can be attributed to its high lipophilicity, which allows it to readily reach the enzymatic active sites, causing functional alterations and inducing oxidative stress [34,35,36]. The doubling of the lag phase by carvacrol at 32 °C suggests that this compound maintains, or even enhances, its efficacy under climatic change scenarios, making it a promising candidate for sustainable control strategies.

Eugenol ranked as the second most effective compound. Although its overall effect was lower than that of carvacrol, eugenol also caused a substantial delay in the onset of exponential growth. This effect may also be attributed to the presence of an aromatic ring supplemented with an OH group capable of forming hydrogen bonds with the active sites of various enzymes [37]. Notably, its antifungal activity remained stable across both temperatures, suggesting a temperature-independent mechanism of action. This stability suggests that formulations based on such bioactive compounds could retain their efficacy even under climate change-driven scenarios involving elevated temperatures, where pest incidence is expected to increase and food safety may be compromised [38].

In contrast, 1-heptyn-3-ol and *trans*-cinnamaldehyde exhibited more modest effects. The inhibitory activity of 1-heptyn-3-ol remained unchanged across concentrations and temperatures, while *trans*-cinnamaldehyde displayed a temperature-dependent behavior. Although the inhibitory effects observed for *trans*-cinnamaldehyde in this study were low, mainly at 28 °C, the antifungal properties of this compound are widely reported [39,40,41,42,43]. Conversely, there are no previous reports on the effects of 1-heptyn-3-ol on the growth of *F. verticillioides*. However, based on previous work on 1-octyn-3-ol [44], a structurally related compound, the compound was expected to exhibit strong antifungal activity due to the presence of a triple bond, which confers nucleophilic properties and makes it susceptible to addition reactions [45,46,47]. Although these reactions may occur inside fungal cells, the inhibitory effect of 1-heptyn-3-ol observed here is low, indicating that this compound is not a suitable candidate for controlling *F. verticillioides*.

Taken together, these findings provide important insights into how projected climate change conditions—namely elevated temperatures—may influence the growth dynamics of *F. verticillioides*, as well as the efficacy of natural antifungal compounds. The stable or enhanced activity of carvacrol and eugenol under simulated climate conditions supports their potential as climate-resilient biocontrol agents. Previous studies have demonstrated the high efficacy of these compounds in inhibiting fumonisin B1 production [34,48]. Accordingly, the results obtained in the present study underscore the need to conduct further experiments to evaluate their performance under more realistic and dynamic environmental conditions.

Moreover, based on the background literature and the effects observed in this work, the design of a mixture combining trans-cinnamaldehyde with more bioactive compounds, such as carvacrol or eugenol, could be proposed to enhance antifungal efficacy and to assess the behavior of this mixture under climate change scenarios.

Collectively, these results reinforce the need for further research to explore the molecular mechanisms underlying these responses and to evaluate the field applicability of compound mixtures, particularly those combining carvacrol or eugenol with less active agents such as *trans*-cinnamaldehyde. Ultimately, the development of natural, climate-resilient formulations may offer a sustainable path forward for integrated strategies aimed at managing phytopathogens and reducing mycotoxin contamination in agricultural systems.

### 3.2. Enhanced Efficacy of 1-Heptyn-3-ol, Carvacrol, Eugenol, and Trans-Cinnamaldehyde Against *S. zeamais* Under Global Warming Scenarios

In the present study, we demonstrate that temperature plays a critical role in modulating the acute insecticidal activity of organic volatile compounds against *S. zeamais*. In both scenarios, mortality did not exceed 5%, indicating that the increase in temperature alone does not affect insect survival. Under control conditions (28 °C), 1-heptyn-3-ol exhibited the highest efficacy among the four compounds tested, producing complete mortality even at intermediate and low concentrations. In contrast, carvacrol, *trans*-cinnamaldehyde, and eugenol showed a clear dose-dependent pattern, with marked reductions in mortality at lower concentrations.

Exposure to a projected global warming scenario (32 °C, RCP 8.5) substantially altered these efficacy patterns. Notably, 1-heptyn-3-ol maintained maximum efficacy at all concentrations, indicating that its insecticidal action is robust to temperature increases. Moreover, carvacrol, *trans*-cinnamaldehyde, and eugenol all displayed enhanced performance at 32 °C compared with 28 °C. In particular, carvacrol reached nearly complete mortality even at reduced concentrations. Although these compounds did not display clear dose–response relationships at 32 °C, the overall increase in mortality suggests that moderate warming may amplify the toxic effects of these natural compounds.

In previous studies, we demonstrated the detrimental effect of 1-heptyn-3-ol against this insect [45]. The reactivity of this compound has been attributed not only to the length of its carbon chain and the presence of the hydroxyl group, but also to the triple bond in its structure [45,46,49,50,51]. The consistently high mortality observed under both temperature scenarios suggests that the elevated reactivity of this compound may be largely independent of temperature.

Similarly, carvacrol, eugenol, and *trans*-cinnamaldehyde are plant-derived bioactive compounds with demonstrated insecticidal activity against *S. zeamais* [52,53,54,55]. Their toxicity is primarily associated with disruptions in feeding and reproductive behavior and inhibition of acetylcholinesterase activity [52,53,54,55,56,57,58,59]. Interestingly, in the present study, an increase in mortality was observed at higher temperatures, a pattern not detected in control treatments, indicating that these compounds may exhibit enhanced bioactivity under projected global warming scenarios. This result is consistent with previous studies reporting that higher temperatures may increase volatilization rates, thereby enhancing insect exposure. Additionally, elevated temperatures may accelerate or modify respiratory rates or increase cuticle permeability in insects [60,61,62]. In this context, future studies incorporating direct measurements of vapor pressure or headspace concentration would help clarify the physicochemical effects associated with volatilization under elevated temperatures.

Overall, these results reinforce previous reports emphasizing the high potency of alkynyl alcohols against stored-product pests, while also confirming the moderate yet variable activity of phenylpropanoids and terpenoids under standard environmental conditions. Also, these findings suggest that the bioactivity of all tested compounds is mediated by multiple biochemical mechanisms, highlighting their potential as natural insecticides under diverse environmental conditions.

From an applied perspective, the finding that natural compounds retain or improve their insecticidal performance at elevated temperatures is particularly relevant, as climate change is expected to intensify pest pressures in stored-grain systems [31,33]. Identifying compounds whose activity is not diminished under warmer conditions is therefore crucial for future pest management strategies. The temperature resilience of 1-heptyn-3-ol and the temperature-enhanced efficacy of carvacrol, *trans*-cinnamaldehyde, and eugenol suggest that these compounds could be integrated into effective and sustainable control tools under warming climate conditions. Further research should investigate the biochemical mechanisms underlying temperature-mediated enhancements in toxicity, assess long-term stability and volatilization kinetics, and evaluate these compounds under realistic storage conditions to validate their suitability for climate-resilient pest management programs. Additionally, it will be essential to examine how the metabolism of successive insect generations exposed to projected global warming temperatures may shift, as potential physiological or metabolic adaptations could alter susceptibility patterns and ultimately influence the long-term effectiveness of these natural compounds.

## 4. Materials and Methods

### 4.1. Fungal Strain, Insects, and Natural Organic Compounds

The toxicogenic strain of *Fusarium verticillioides* (Sacc.) Nirenberg (M3125) [63] was obtained from the Department of Agriculture, Agricultural Research Service, National Center for Agricultural Utilization Research (Peoria, IL, USA), provided by Dr. Robert Proctor. For bioassays, conidial suspensions (1 × 10^6^ CFU/mL) were prepared according to Dambolena et al., (2008) [64].

Adults of *Sitophilus zeamais* (Motschulsky) were reared under storage conditions on insecticide-free maize at 28 ± 2 °C and 70 ± 5% relative humidity [65].

The organic compounds used in this study were 5-Isopropyl-2-methylphenol (carvacrol, 98% purity), 1-heptyn-3-ol (97%), 2-methoxy-4-(2-propenyl)-phenol (eugenol, ≥98%), and trans-3-phenyl-2-propenal (*trans*-cinnamaldehyde, ≥99%) (Figure 3). Given that the pesticidal effectiveness of the natural organic compounds carvacrol, eugenol, and *trans*-cinnamaldehyde has been widely documented [42,48,52,54,66], these compounds were selected to evaluate their activity under simulated global warming scenarios. Although 1-heptyn-3-ol is a synthetic compound, it was included in this study due to the lack of reports on its activity against *F. verticillioides*. Previous findings have shown its insecticidal activity against *Sitophilus zeamais* and no phytotoxic effects on *Zea mays* seeds, supporting its potential as a candidate for integrated pest management strategies [45]. All compounds were purchased from Merck^®^ (Buenos Aires, Argentina).

### 4.2. Climate Change Scenarios

To assess the impact of climate change on the development of *F. verticillioides* and the survival of *S. zeamais*, two temperature scenarios were considered. The temperatures to be tested were selected considering the RCP 8.5 scenario (3.5 °C for the period 2081–2100) in the central region of Argentina [11]. The control temperature corresponds to the average temperature during the maize sowing period (2010–2023) [67]. Accordingly, the experimental temperatures were set at 28 ± 1 °C and 32 ± 1 °C, control and treatment, respectively.

### 4.3. Effect of Organic Compounds on Fungal Growth

To determine the effects of natural organic compounds and 1-heptyn-3-ol on the vegetative growth of *F. verticillioides* the Minimum Inhibitory Concentration (MIC) technique was applied [68]. Briefly, the selected compounds were incorporated into Potato Dextrose Agar (PDA) at increasing concentrations (0, 0.25, 0.5, and 1 ppm) and poured into sterile Petri dishes (Ø = 9.0 cm). Each Petri dish was inoculated with 5 µL of conidial suspension (1 × 10^6^ CFU/mL) and incubated as follows: three replicates of each concentration/compound were maintained at either 28 ± 1 °C or 32 ± 1 °C (temperature treatment).

For all replicates, mycelial diameter was measured daily until the control plates (0 ppm) were completely colonized. Lag phase and Inhibition percentage were subsequently calculated.

### 4.4. Insecticidal Effect of Organic Compounds Against S. zeamais

The insecticidal effect of the organic compounds under climate change conditions was evaluated following the fumigant toxicity protocol described by Herrera et al. (2015) [69]. Three concentrations of each compound (0.125, 0.025, and 0.0125 ppm) were applied to filter paper disks (Ø = 2 cm) and placed on the inner side of the lids of 30 mL amber glass flasks. Ten *S. zeamais* adults were then introduced into each flask and incubated under the corresponding environmental conditions. A nylon gauze was used to prevent direct contact between the insects and the compounds. Mortality was recorded after 24 h.

### 4.5. Statistical Analysis

For growth data analysis, linear regression was performed for each concentration and atmospheric condition to calculate growth rates (mm/h). Only the linear portion of each growth curve was used, with adjusted R^2^ values ranging from 0.90 to 1 (n_temperature_ = 33). Subsequently, lag phase values were calculated and compared using analysis of variance (ANOVA). When statistically significant differences were obtained (*p*-value ANOVA ≤ 0.05), a post hoc DGC test was applied.

Mortality data for the tested compounds were analyzed using ANOVA followed by a DGC post hoc test (ANOVA, *p* ≤ 0.05). For each concentration, at least five replicates were performed.

For all statistical analyses, assumptions of normality and homogeneity of variance were verified (*p* ≥ 0.05). All statistical analyses and graphical representations were performed using Navure software (Professional+ v 3.0) (Córdoba, Argentina) [70].

## 5. Conclusions

The present study demonstrates that both fungal and insect targets respond differently to temperature increases, yet natural bioactive compounds maintain or enhance their efficacy under elevated temperatures. Carvacrol and eugenol exhibited sustained antifungal activity against *F. verticillioides*, significantly delaying growth even under simulated climate change conditions. In contrast, 1-heptyn-3-ol, carvacrol, eugenol, and *trans*-cinnamaldehyde retained or improved their insecticidal activity against *S. zeamais* at higher temperatures.

Importantly, the identification of multifunctional compounds capable of controlling both fungal pathogens and insect pests is particularly relevant in stored-grain systems, where infestation by *S. zeamais* not only causes direct losses but also promotes fungal development by increasing grain damage and local humidity [20]. In this context, the dual efficacy of carvacrol highlights its potential as a broad-spectrum agent capable of mitigating two interacting threats.

Moreover, the development of formulations or compound mixtures with strong antifungal and insecticidal activity represents a promising avenue for future research, particularly under global warming scenarios. In addition, elucidating the biochemical mechanisms and physiological processes underlying the effectiveness of these natural compounds under increased temperature conditions constitutes an important next step.

The current agricultural production model is based on the construction of deeply simplified agroecosystems, in which ecosystem services are often overlooked and which depend on the constant input of external resources. Such systems are prone to rapid degradation due to biodiversity loss and deterioration of environmental services [71,72]. The use of natural bioactive compounds also offers significant environmental advantages. Their low persistence in the environment and reduced risk to non-target organisms make them safer alternatives to conventional synthetic pesticides [73,74,75,76,77].

Therefore, the results presented in this study provide valuable information for the development of rational and environmentally friendly pest control strategies that account for potential changes in pest aggressiveness in a context of global warming. Future studies should evaluate the effect of these strategies on non-target organisms under projected climate conditions.

Taken together, these findings underscore the importance of advancing both the study and application of natural bioactive compounds as sustainable and climate-resilient tools for protecting crop health and food safety in the face of climate change. Future research should also investigate how rising atmospheric CO_2_ concentrations—another major driver of global change—may influence pest physiology, population dynamics, and the performance of natural compounds, as such interactions could further shape the efficacy of integrated control strategies under projected environmental conditions.

## Figures and Tables

**Figure 1 plants-15-00048-f001:**
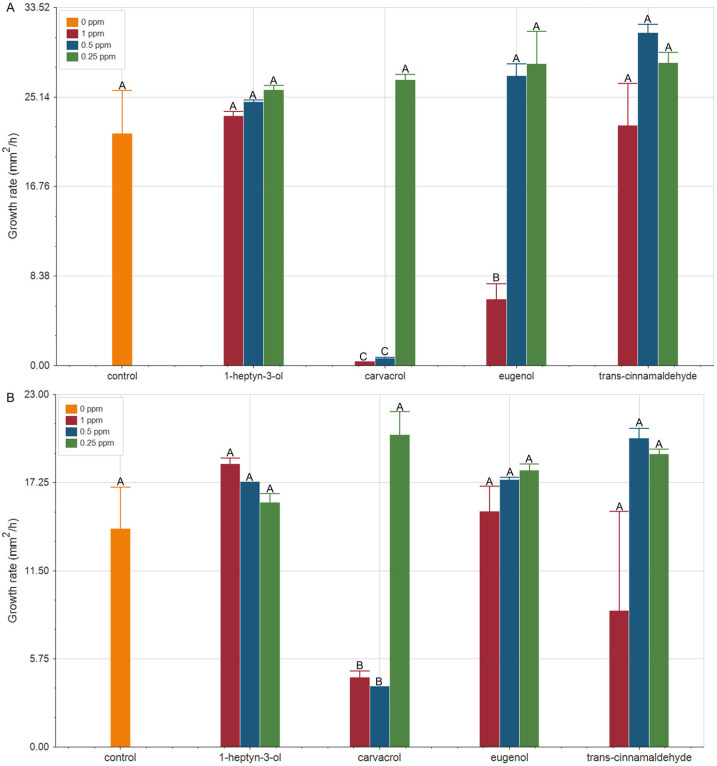
Growth rate (mm^2^/h) of *F. verticillioides* exposed to increasing concentrations of carvacrol, eugenol, *trans*-cinnamaldehyde, and 1-heptyn-3-ol under two global warming scenarios: (**A**) 28 °C (*p* < 0.0001, df = 12; F = 55.70) and (**B**) 32 °C (*p* < 0.0001, df = 12; F = 8.34). Green bars: 0.25 ppm; Blue bars: 0.5 ppm; Red bars: 1 ppm; Orange bars: 0 ppm (controls). Different letters indicate statistically significant differences according to ANOVA followed by a post hoc DGC test.

**Figure 2 plants-15-00048-f002:**
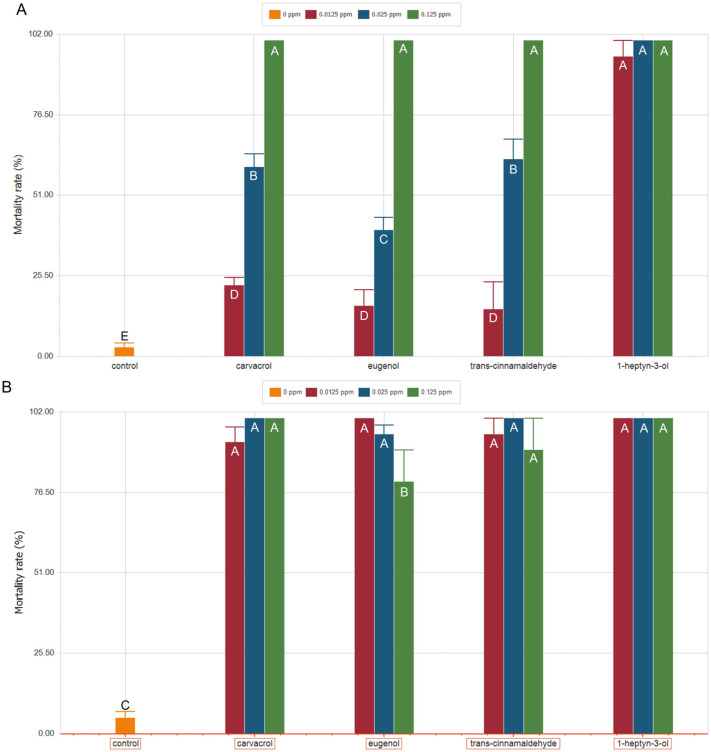
Mortality rate (%) of *S. zeamais* after 24 h of exposure to increasing concentrations of carvacrol, eugenol, *trans*-cinnamaldehyde, and 1-heptyn-3-ol under two global warming scenarios: (**A**) 28 °C (*p* < 0.0001, df = 12; F = 98.4) and (**B**) 32 °C (*p* < 0.0001, df = 12; F = 154.6). Green bars: 0.125 ppm; Blue bars: 0.025 ppm; Red bars: 0.0125 ppm; Orange bars: 0 ppm (controls). Different letters indicate statistically significant differences according to ANOVA followed by a post hoc DGC test.

**Figure 3 plants-15-00048-f003:**
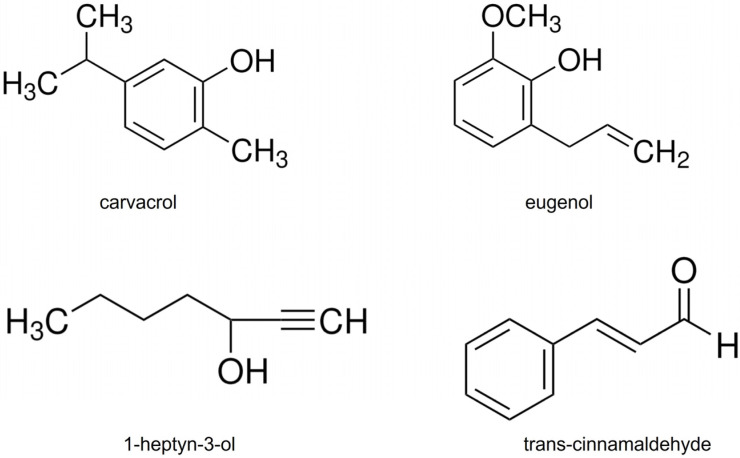
Molecular structure of selected organic compounds.

## Data Availability

The raw data supporting the conclusions of this article will be made available by the authors upon request (vusseglio@imbiv.unc.edu.ar).

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
