# Peer review of "Organic Compounds as a Natural Alternative for Pest Control: How Will Climate Change Affect Their Effectiveness?"

_plants, 2025, doi:10.3390/plants15010048_

Round 1

Reviewer 1 Report

Comments and Suggestions for Authors

I have carefully reviewed the manuscript “Organic Compounds as a Natural Alternative for Pest Control: How Will Climate Change Affect Their Effectiveness” and it is based on an investigation of the effects of four natural organic compounds—carvacrol, eugenol, trans-cinnamaldehyde, and 1-heptyn-3-ol—on the growth of the fungal pathogens

I have the following suggestions that can be helpful in the overall improvement of this manuscript.

  1. There is limited discussion on the molecular mechanisms underlying the observed effects. For instance, the study does not explore how temperature modulates the biochemical activity of the compounds at a cellular level. Please check
  2. The manuscript reports some data inconsistencies in terms of statistical analysis and figure labeling. For example, in the mortality rate data for zeamais (Figure 2), the text mentions 100% mortality at certain concentrations but doesn't consistently explain the experimental conditions (e.g., exposure duration or concentration ranges). Clarifying these details would improve the reproducibility and transparency of the study.
  3. I would recommend to add some discussion on assessing their safety to beneficial organisms (e.g., pollinators, soil microbiota). Adding non-target effects of these compounds would strengthen the manuscript's argument for their use in sustainable agriculture.
  4. Ensure consistency in data reporting and figure labeling, and provide clearer explanations for the experimental conditions

Author Response

Response to Reviewer Comments

Thank you for sending the comments with respect to our manuscript entitled “Organic compounds as a natural alternative for the pests control: How will climate change affect their effectiveness?”, now “Organic compounds as a natural alternative for pest control: How will climate change affect their effectiveness?”. We appreciate the comments made by the reviewer that will help to improve the manuscript. We have improved the manuscript taking into account the suggestions and comments of the referee. We have highlighted the changes performed in the modified version of the manuscript, in yellow. Also, the language and wording were reviewed by a professional translator.

Comments and Suggestions for Authors

Reviewer #1

I have carefully reviewed the manuscript “Organic Compounds as a Natural Alternative for Pest Control: How Will Climate Change Affect Their Effectiveness” and it is based on an investigation of the effects of four natural organic compounds—carvacrol, eugenol, trans-cinnamaldehyde, and 1-heptyn-3-ol—on the growth of the fungal pathogens

I have the following suggestions that can be helpful in the overall improvement of this manuscript.

  • There is limited discussion on the molecular mechanisms underlying the observed effects. For instance, the study does not explore how temperature modulates the biochemical activity of the compounds at a cellular level. Please check

Response: Thank you for your observation. We agree that exploring the molecular and cellular mechanisms underlying the observed effects—and particularly how temperature may modulate the biochemical activity of these natural compounds—is an important and compelling research direction. Although this aspect was beyond the scope of the present study, we have now incorporated a statement in the revised manuscript indicating that temperature-dependent biochemical processes at the cellular level will be considered in future work. Specifically, we added a sentence noting that the biochemical pathways modulated by temperature represent an important next step to be evaluated in subsequent studies. (Lines 337-341)

  • The manuscript reports some data inconsistencies in terms of statistical analysis and figure labeling. For example, in the mortality rate data for zeamais (Figure 2), the text mentions 100% mortality at certain concentrations but doesn't consistently explain the experimental conditions (e.g., exposure duration or concentration ranges). Clarifying these details would improve the reproducibility and transparency of the study.

Response: Thank you for your recommendation. We have incorporated the necessary clarifications in the manuscript to ensure consistency between the statistical analysis, figure labeling, and the experimental conditions described in the text. For Figures 1 and 2, we added detailed information in the captions regarding the tested concentrations and the corresponding bar colors.

Regarding the mortality assays against S. zeamais, we clarified that these were acute insecticidal assays, in which insects were exposed for 24 h to the tested concentrations (0.125, 0.025, and 0.0125 ppm). Mortality rates were calculated based solely on this exposure period. These details have been explicitly included in both the Materials and Methods and Results sections (Lines 106–144 and 301–309) to enhance reproducibility and transparency.

  • I would recommend to add some discussion on assessing their safety to beneficial organisms (e.g., pollinators, soil microbiota). Adding non-target effects of these compounds would strengthen the manuscript's argument for their use in sustainable agriculture.

Response: Thank you for your suggestion. In the conclusions section we have included a brief reflection on the benefits of using VOCs in agroecosystems. (Lines 342-348)

  • Ensure consistency in data reporting and figure labeling, and provide clearer explanations for the experimental conditions.

Response: Thank you for your observation. The figures have been carefully reviewed, and additional information regarding the experimental conditions has been incorporated into the figure captions, the Results section, and the Materials and Methods section.

Reviewer 2 Report

Comments and Suggestions for Authors

The manuscript by Usseglio et al investigates the antifungal and insecticidal activity of four natural organic compounds under two temperature scenarios representing current conditions and projected global warming. The study addresses an important and timely topic—how climate change may alter the performance of natural biopesticides. The experiments are well described, and the results are presented clearly. The work offers valuable insights into the temperature-dependent activity of these compounds.
However, several issues should be addressed to make the manuscript suitable for publication. Below, the comments of this Reviewer:

- the Authors should explain better the choice of the compounds (three natural on one synthetic) assessed in the manuscript;
- the insecticidal effects are based on fumigation. Temperature strongly affects vapor pressure. Higher mortality at 32 °C may result from increased volatilization rather than inherent biological sensitivity. The Authors shoul measure  vapor pressure or headspace concentration of each compound at 28 °C and 32 °C (GC-MS recommended) for clarifying the  chemical–physical effects (volatilization) from biological effects;
- discussion invokes mechanisms like ROS generation, membrane disruption, and enzyme inhibition but does not measure them. Please, support the discussion with experimental measures;
-the discussion proposes mixtures but no experiments test them. Please, provide experimental data to support or rephrase the sentence.

Comments on the Quality of English Language

The English writing is generally clear but a polishing by English editor could be beneficial.

Author Response

Response to Reviewer Comments

Thank you for sending the comments with respect to our manuscript entitled “Organic compounds as a natural alternative for the pests control: How will climate change affect their effectiveness?”, now “Organic compounds as a natural alternative for pest control: How will climate change affect their effectiveness?”. We appreciate the comments made by the reviewer that will help to improve the manuscript. We have improved the manuscript taking into account the suggestions and comments of the referee. We have highlighted the changes performed in the modified version of the manuscript, in yellow. Also, the language and wording were reviewed by a professional translator.

Comments and Suggestions for Authors

Reviewer #2

The manuscript by Usseglio et al investigates the antifungal and insecticidal activity of four natural organic compounds under two temperature scenarios representing current conditions and projected global warming. The study addresses an important and timely topic—how climate change may alter the performance of natural biopesticides. The experiments are well described, and the results are presented clearly. The work offers valuable insights into the temperature-dependent activity of these compounds.

However, several issues should be addressed to make the manuscript suitable for publication. Below, the comments of this Reviewer:

  • the Authors should explain better the choice of the compounds (three natural on one synthetic) assessed in the manuscript;

Response: Thank you for your suggestion. Given that the pesticidal effectiveness of the natural organic compounds carvacrol, eugenol, and trans-cinnamaldehyde is widely documented[1–5], these compounds were selected to evaluate their activity under simulated global warming scenarios Although 1-heptyn-3-ol is a synthetic compound, it was included in this study due to the lack of reports on its activity against this fungus. Previous findings have shown its insecticidal activity against Sitophilus zeamais and no phytotoxic effects on Zea mays seeds[6], supporting its potential as a candidate for inte-grated pest management strategies. These clarifications were included into the manuscript (Lines 270-280)

  • the insecticidal effects are based on fumigation. Temperature strongly affects vapor pressure. Higher mortality at 32 °C may result from increased volatilization rather than inherent biological sensitivity. The Authors shoul measure vapor pressure or headspace concentration of each compound at 28 °C and 32 °C (GC-MS recommended) for clarifying the chemical–physical effects (volatilization) from biological effects;

Response: Thank you for your observations. We agree that the ideal approach would involve experimentally comparing the headspace concentrations of each compound at both working temperatures. However, due to time constraints, it was not possible to perform these analyses within the current study.

In this case, all compounds evaluated exhibit relatively low vapor pressures. Therefore, it is expected that an increase in temperature would enhance their volatilization, resulting in higher concentrations in the headspace and making the compounds more readily available for insect uptake, which could in turn contribute to the higher mortality observed at 32 °C. This explanation is consistent with the behavior of low–vapor-pressure compounds and has been added in the discussion.

We agree that future work should incorporate direct measurements of vapor pressure or headspace concentration (e.g., via GC–MS) to more clearly distinguish the chemical–physical effects related to volatilization from the biological sensitivity of the insects. The manuscript has been revised to reflect this point. (Lines 236-239)

  • discussion invokes mechanisms like ROS generation, membrane disruption, and enzyme inhibition but does not measure them. Please, support the discussion with experimental measures;

Response: Thank you for this observation. In response, we have revised the manuscript to remove statements referencing potential mechanisms of action (e.g., ROS generation, membrane disruption, and enzyme inhibition) that other authors have previously reported but which were not experimentally evaluated in our study. The discussion has been adjusted to strictly reflect our results, avoiding mechanistic interpretations of phenomena we did not directly measure.

We believe that understanding the biochemical mechanisms underlying the observed efficacy of natural compounds is an important step. Therefore, we have added a statement to the revised manuscript indicating that future research will focus on experimentally evaluating how temperature and natural bioactive compounds influence biochemical pathways and physiological processes in the target organisms of pests, which could explain their inhibitory effects. (Lines 337-341)

  • the discussion proposes mixtures but no experiments test them. Please, provide experimental data to support or rephrase the sentence.

Response: Thank you for your suggestion. The sentence has been restructured to clarify that our intention was not to imply that mixture formulations were experimentally evaluated in the present study. Rather, we wanted to propose, as future objectives, the development of mixtures that could improve the efficacy of low-activity compounds, broaden their potential sites of action, and help delay the development of resistance. In this context, such multi-target formulations could offer the advantage of simultaneously attacking fungal and insecticidal pests that affect stored corn. The revised text now reflects this forward-looking perspective without suggesting that mixture experiments were conducted in this work. (Lines 337-341)

References

  1. Zaio, Y.P.; Gatti, G.; Ponce, A.; Saavedra Larralde, N.A.; Martinez, M.J.; Zunino, M.P.; Zygadlo, J.A. Cinnamaldehyde and Related Phenylpropanoids, Natural Repellents, and Insecticides against Sitophilus Zeamais (Motsch.). A Chemical Structure-Bioactivity Relationship. J. Sci. Food Agric. 2018, 98, 5822–5831, doi:10.1002/jsfa.9132.
  2. Rodríguez, A.; Beato, M.; Usseglio, V.L.; Camina, J.; Zygadlo, J.A.; Dambolena, J.S.; Zunino, M.P. Phenolic Compounds as Controllers of Sitophilus Zeamais: A Look at the Structure-Activity Relationship. J. Stored Prod. Res. 2022, 99, 1–8, doi:10.1016/j.jspr.2022.102038.
  3. Dambolena, J.S.; Zygadlo, J.A.; Rubinstein, H.R. Antifumonisin Activity of Natural Phenolic Compounds. A Structure-Property-Activity Relationship Study. Int. J. Food Microbiol. 2011, 145, 140–146, doi:10.1016/j.ijfoodmicro.2010.12.001.
  4. Xie, Y.; Huang, Q.; Wang, Z.; Cao, H.; Zhang, D. Structure-Activity Relationships of Cinnamaldehyde and Eugenol Derivatives against Plant Pathogenic Fungi. Ind. Crops Prod. 2017, 97, 388–394, doi:10.1016/j.indcrop.2016.12.043.
  5. Peng, S.; Zhang, H.; Xu, R.; Chang, X.; Zhou, Z.; Yang, Y.; Xiang, H.; Li, Y. Inhibitory Effect of Eugenol on Fusarium Oxysporum and Transcriptome Analysis. Tob. Sci. Technol. 2025, 58, 38, doi:10.16135/j.issn1002-0861.2024.0938.
  6. Cano, M.C.; Beato, M.; Usseglio, V.L.; Merlo, C.; Zunino, M.P. Bioactive Paints with Volatile Organic Alcohols for the Control of Sitophilus Zeamais. J. Stored Prod. Res. 2024, 109, 102423, doi:10.1016/j.jspr.2024.102423.

Round 2

Reviewer 1 Report

Comments and Suggestions for Authors

I would like to recommend the manuscript for publication since the authors have incorporated all my concerns

Reviewer 2 Report

Comments and Suggestions for Authors

Authors have reply satisfactorily to all comments.